# REPRESENTING SENTENCE STRUCTURE IN A TREE METRIC SPACE

## ABSTRACT

This paper proposes building a sentence tree metric space through representation learning of sentence structure. Our method represents every sentence tree structure as a vector, with the Euclidean distance applied to construct the sentence tree metric. In comparison with the previous, representative tree-metric methods of the (tree edit distance) TED, tree kernels, and PQ-grams, our method has the best computational complexity, scaling to handle a million trees, yet it performs well in predicting tree structure and learning TED-like distances, even without TED for supervision. Our large-scale sentence metric space analyses provide novel ways to study sentence structures from recent language technology, by evaluating parsers and tree-annotated corpora, and with tree structures acquired by recent large language models (LLMs). These analyses also address the nature of natural language trees not only within languages but in comparison with random trees.

## 1 INTRODUCTION

Large language models (LLMs) have shown the capability to process sentences with long dependence and are acquired without being trained on annotated sentence structure. However, the acquired structure remains in a black box without explicit understanding of SOTA technology regarding tree structure, which limits the potential improvement.

To tackle this problem, we propose a new representation learning of sentence tree structure by using Transformers (Vaswani et al., 2017). As Transformers have been effective in reproducing long dependence, we used them to build a model dedicated to tree representation in a metric space. For a set of trees $T$, our method, trains a function $f$ that produces $h = f(t)$ for $t \in T$, where $h \in \mathcal{R}^H$, a vector of hyperparameter dimension $H$. For $h_1, h_2 \in \mathcal{R}^H$, corresponding to $t_1 \in T_1$ and $t_2 \in T_2$, respectively, the Euclidean distance in $\mathcal{R}^H$ forms a metric space. With such a space, we can analyze the proximity of two sets of trees, $T_1, T_2$; for example, we can analyze how close French or Japanese is to English. Further, we can discover how biased natural language trees are in comparison with diverse random trees, including context-free trees and even ChatGPT trees. To the best of our knowledge, this approach of representation learning for sentence trees by using deep learning to construct a metric space is novel.

The largest benefit of our approach in comparison with previous work on tree metrics lies in the computational complexity for distance calculation among trees. The tree edit distance (TED) (Tai, 1979) has seen the most application as a representative, fundamental metric among trees, and the mathematical nature of its space has been studied (Feragen et al., 2013). Even with the most efficient algorithm ZSS (Zhang & Shasha, 1989), however, the calculation for trees of size $n$ is $O(n^3)$, with a limitation on speed-up because of its recursive nature. This limits comparison up to hundreds of trees, at best. In contrast, our tree representation enables comparison among a million trees, allowing for a large scale analysis of tree shape differences.

Building such a scalable tree metric space enables quality evaluation at the technology frontier with respect to tree structure. For example, we can understand the limitation of ChatGPT-generated sentence tree structures, or of parser and treebank qualities. Furthermore, we can obtain new understandings of natural language trees. For example, our metric space sheds light on linguistic questions of how biased natural language trees are, regarding "context-freeness" (Shieber, 1985). Our new evidence could enable further improvement in such SOTA technology.

## 2 RELATED WORK

### 2.1 TREE REPRESENTATION AND ITS LEARNING

A tree metric space, built for a set of trees $T$ with a metric function $m$ among elements of $T$, was originally mathematically considered for phylogenetic trees (Billera et al., 2001). Feragen et al. (2013) studied the mathematical na-

Table 1: Qualitative comparison of methods that can train on trees

| Method | tree represen- tation | metric space | speed | | tree/ class prediction |
|--------|--------|--------|--------|--------|--------|
| | | | distance | represen- tation | |
| TEDs | no | yes | $O(n^3)$ | — | no |
| Tree kernels | inexplicit | yes | $O(n^2)$ | — | no |
| PQ-grams | profile | pseudo | $O(n)$ | $O(n)$ | no |
| Tree-LSTM | no | no | — | — | yes |
| Tree-Transformer | no | no | — | — | yes |
| Ours | vector | yes | $O(n)$ | $O(n)$ | yes |

ture of the space generated by TED and its extensions. To the best of our knowledge, there have been no further analyses of metric spaces when they learn data.

There are, however, diverse machine learning methods that train on trees for engineering perspectives. There are roughly two kinds of methods, as summarized in the first two blocks of Table 1, with our proposal in the last row. The first block lists tree metrics and their learning. Early studies of learning metric functions originated in phylogenetic trees, too (Buneman, 1971; 1974). These metrics are acquired through machine learning as originally attempted by Bernard et al. (2006); Boyer et al. (2007). Every tree metric and representation since involves machine learning.

TED, in the first line, is the most fundamental metric (Tai, 1979) but has another limitation besides the complexity mentioned above: the design of the edit costs. Paaßen et al. (2018) proposed to train the costs by embedding the tree nodes in the vector space, but that method stays within the TED framework. Tree kernels, in the second line, represent a tree via all subtrees (Collins & Duffy, 2001; Moschitti, 2004; 2006; Moschitti et al., 2008). This exponential tree representation remains inexplicit and is only tractable via tree kernels as the metric between trees. In the third line are PQ-grams (Augsten et al., 2005), a unique representation of trees through a subtree profile, with a training method proposed by Shindo et al. (2020).

All of these methods form tree metric spaces (third column, Table 1), but the distance calculations for tree kernels and TED are costly (fourth). Furthermore, with no other add-ons, they remain in the metric framework without the predictive capabilities of tree structure or set kinds (last column).

Another large category of previous work entails prediction models built to incorporate/analyze sentence structure, as in the second block of Table 1). There have been proposals to build sequence models incorporating tree structure for a hidden Markov model (Collins, 2002) and an LSTM (Tai et al., 2015). Tree-Transformer (Yang & Dredze, 2016; Wang et al., 2019) is dedicated to grammar induction. These methods target diverse predictions (last column), but they don't represent trees (second) and thus don't involve tree metric spaces (third).

Our motivation lies in using prediction models to build a tree metric space. The last row indicates our model's distinctiveness: it provides a tree representation tree via a vector, with the great benefit of low computational complexity, enabling tree metric analysis among a million trees.

There are broader approaches to graph embedding methods (Narayanan et al., 2017) and to graph neural networks applied to trees. We focus on tree-dedicated models because standard graph methods, especially GNNs without explicit structural encodings, are reported to fail in capturing tree topology and geometry Zhang et al. (2024). This latter work, however, is specifically dedicated to geometric tree classification and is therefore different from ours.

### 2.2 ANALYSES OF SENTENCE STRUCTURE

The two representative formalizations of natural language syntax are Chomskian generative grammar (Chomsky, 1957) and dependency grammar (Tesnière, 1959; Rondal, 1988). Both approaches generated abundant resources of syntactically annotated data, starting with the Penn Treebank (Marcus et al., 1993). In this work, we build our model with universal dependency (UD) tree structure (Nivre et al., 2020), because of its abundant data. Our method also provides new means to assess the quality of UD trees, as will be argued at the end.

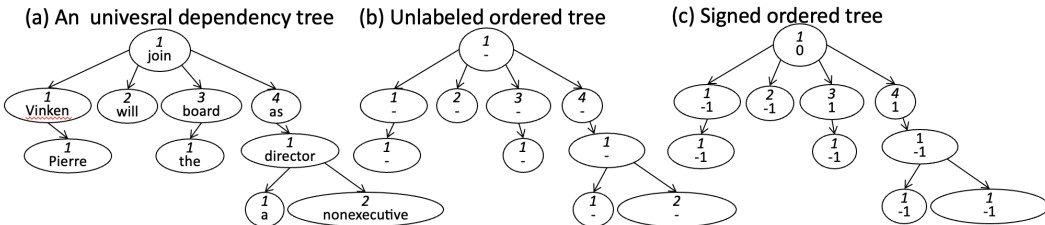

Figure 1: Dependency structure examples of (a) a sentence's dependency tree, and its (b) ordered unlabeled tree and (c) signed ordered tree.

Treebanks have been used in studies on the nature of sentence structure, partly in the field of quantitative linguistics and mainly for modifier-modified relations: for dependency length (Liu et al., 2017), assuming optimality of the "minimal" length; and for dependency direction (Liu, 2010), which has a linguistic background in Greenberg universals (Greenberg, 1963; Dryer, 1992). More recently, using structural probing, Hewitt & Manning (2019); Chi et al. (2020); Xu et al. (2022) analyzed modifier-modified relations of syntax represented in BERT. In contrast, this work considers the characteristics of the entire tree shape and compares trees accordingly, along with random trees.

Reverse engineering of tree structure from sentences, i.e., parsing, as studied in the natural language processing field, is typically implemented through supervision of these treebanks. However, unsupervised parsing that does not require a parser or a treebank is achieved by LLMs. Although this work has a limited relation with parsing techniques themselves, in section 6.1 we show that the proposed method serves to assess parsing quality, and that it provides evidence that LLMs achieve SOTA parser quality.

Further, our method provides new means to study the fundamental nature of natural language trees. One scientific question we consider in this work is the question of the "context-freeness" of natural language (Chomsky, 1956; 1959). Previously, natural language has been reported to be beyond context free, as in Shieber (1985).

## 3 SENTENCE TREE METRIC SPACE

### 3.1 SENTENCE TREES

Let $t = (V, E)$ denote a rooted ordered directed tree (also known as a plane tree), where $V$ is the set of nodes and $E \subset V \times V$ of edges, with ordering specified for the children of each node. Let $T$ denote a set of finite rooted ordered directed trees. Let $n$ be the number of nodes for a tree $t \in T$.

This work considers UD structures (Nivre et al., 2020) as $T$, although our representation applies also to phrase structure. In UD, the tree root is the sentence head, and an edge represents a relation between two words, from the head to one of its modifiers. Figure 1(**a**) shows an example of size $n = 10$. A node corresponds to a word, and a branch runs from the head to each of its modifiers. The italicized numbers in each node give the order among siblings.

Among multiple UD tree options deriving from this original dependency structure, the most basic when considering random trees is **unlabeled ordered trees**, where the words are anonymized, as shown in Fig. 1(**b**) with all words denoted by '_'. Furthermore, we may consider a variation shown in Fig. 1(**c**). In each sentence, a child's position is indicated by -1 or 1 when it is before or after the parent, respectively. The latter has an advantage in that a tree structure is transformable into a unique sequence under the constraint of deeper nodes having closer modified-modifier placement.

Another possibility to consider is to whether to use words or parts of speech (POSs) as nodes. Using words, however, disables comparison among languages, while POSs also vary across languages. Furthermore, the use of words and POSs limits comparison with random trees lacking actual words. Hence, in this work, *our tree representation focuses only on the tree shape*, while an approach incorporating words and POSs remains our future work. The difference between (b) and (c) lies in whether the tree is sequentializable or not. Apart from language, many real world data are represented in nonnumeric symbolic sequences with latent tree structure, such as DNA and music. Although we do not go beyond language in this paper, given this extensibility, we focus on structure (c). In other words, unlike all the linguistic work mentioned in section 2.2, our work does *not use*

*words or POSs* but solely considers tree structure. We will show that, even then, languages can be characterized within and beyond.

## 3.2 METRIC BETWEEN TREES

A metric space is a set $T$ together with a distance function $m$ : $T \times T \to [0, \infty)$. To form a metric, $m$ must satisfy nonnegativity, identity of indiscernibles, symmetry, and the triangle inequality; many similarity functions e.g. the vector cosine similarity do not fulfill.

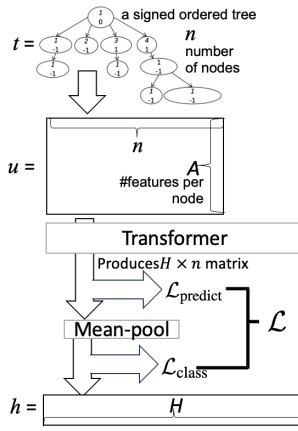

We want to design a simple tree metric space that is trainable on sentence trees. We propose a tree representation via a vector $h = f(t)$ with $t \in T$, enabling tree measurement with the *Euclidean* distance and naturally forming the metric: $m(h_1, h_2) = \|h_2 - h_1\|_2$.

We obtain $h$ as shown in Fig. 2. Each node is represented by a feature vector and the tree is transformed into a matrix $u = g(t)$, in the order of depth-first traversal. While the node position in a sequence could be arbitrary, depth-first ordering represents a tree structure well (Vinyals et al., 2015). Hence, $u \in \mathbb{R}^{A \times n}$, where $n$ is the tree size, and $A$ is the number of per-node features (defined in Supplementary B). A Transformer encodes $u$ into node hidden states of an $H \times n$ matrix for a hyperparameter $H$, and the final tree representation $h = f(t) \in \mathbb{R}^H$ is obtained by *mean pooling*. To train on $T$, $u$ is padded to a maximum length $N_{\max}$ among $T$, so that the same formulation applies to all sentences.

Figure 2: Our process of training a tree representation $h = f(t)$.

The function $f(\cdot)$ is trained using the following loss function:

$$\mathcal{L} = \mathcal{L}_{\text{predict}} + \alpha \mathcal{L}_{\text{class}},$$

where $\alpha$ is a hyperparameter. The model thus has two hyperparameters, $\alpha$ and $H$, whose values will be decided in section 5.1. The two terms on the right side are calculated as the *cross-entropy* of the prediction probabilities obtained as $z = \text{softmax}(Wh' + b)$.

Here, $\mathcal{L}_{\text{predict}}$ is the sum of the cross-entropies of the positions of every node in the sentence and those of its parents. $h'$ is the node's corresponding vector in the Transformer's directly output $H \times n$ matrix. On the other hand, $\mathcal{L}_{\text{class}}$ is a tree-level cross-entropy predicting the class (i.e., the language, see section 4.1); therefore, $h' = h$ for this class prediction.

We consider this the minimal model to produce a tree metric space. Among neural architectures, the Transformer has proven its effectiveness in capturing long dependence, better than the RNN architecture and its extensions. Recent Mamba models (Liu et al., 2022) are designed to reproduce longer dependence. As for the loss function design, we want a metric space that can judge similar sets as close even when they are from different sets. Therefore, models that directly separate trees of different classes, such as contrastive learning (van den Oord et al., 2018), are inadequate.

For the metric to measure the distance between two *sets* of trees, we use the Wasserstein distance (Villani, 2008; Peyré & Cuturi, 2019), the SOTA metric among point clouds. It is calculated over the distribution of Euclidean distances between all pairs of trees in the sets. Supplementary A gives the definition and the rationale among various possibilities.

## 4 DATASET AND SETTINGS

### 4.1 SENTENCE DATASET

Table 2 lists the main data we analyzed. In this work, a *class* refers to a tree corpus. This section explains the data in the first and second columns, while the distances given in the third and fourth columns will be discussed in the experiment Sections 5 and 6. As for natural language sentences, this work uses the UD dataset (Nivre et al., 2020), specifically v. 2.15.

The first block of Table 2 summarizes the natural language sentences we used from UD. The principal corpus in this paper is always English-EWT, a collection of written text acquired from English

Web data. Other languages follow, one corpus per language, under the criteria of (i) containing more than 15,000 sentences (required for training and sampling, discussed ahead) of (ii) written text, comprising (iii) non-translated content. Overall, we used 21 languages including English.

The second block lists treebanks obtained by reparsing English-EWT sentences with the SOTA parsing methods Spacy, UDPipe, and ChatGPT4o and 5, which we address later in section 6.1. The last two blocks list treebanks of random trees, as explained in the following section.

### 4.2 RANDOM TREES

Comparison with random trees other than natural language serves to illuminate the nature of natural language tree structure. We generated three kinds of signed ordered random trees as follows. All tree data generated by random models are prefixed by "Rand-" in this paper, as in the last block of Table 2.

**Basic random trees** appear first in the "Random Trees" block. For a tree of length $n$, there are $(n-1)!$ unlabeled trees, each of which is further randomized with node labels. We consider representatives of star, balanced, uniform, and Markov (including linear) trees, as defined in Supplementary C.

**Random Context-Free Trees:** A tree is *context free* in this work if a node samples children *independently* of its siblings (if any exist). We will compare natural language trees with context-free trees in section 6.2. Using the method mentioned in Supplementary D, for each language corpus

Table 2: Treebanks used in this work, listing the #trees and mean Wasserstein distances *from English-EWT* for TED and Ours, obtained through $I = 10$ trials.

| treebank | # trees | TED | Ours |
|---|---|---|---|
| Natural Language | | | |
| English-EWT | 16621 | 2.984±0.173 | 2.417±0.080 |
| Arabic-PADT | 19738 | 7.087±0.044 | 6.439±0.104 |
| Catalan-AnCora | 16678 | 5.145±0.069 | 5.367±0.097 |
| Czech-PDT | 87913 | 5.557±0.066 | 3.367±0.114 |
| Dutch-LassySmall | 17120 | 5.178±0.046 | 3.161±0.065 |
| Estonian-EDT | 30972 | 5.771±0.051 | 3.645±0.076 |
| Finnish-FTB | 18723 | 6.789±0.034 | 4.956±0.062 |
| French-GSD | 18535 | 5.001±0.070 | 4.683 ±0.081 |
| German-HDT | 189928 | 5.300±0.099 | 4.041 ±0.073 |
| Hindi-HDTB | 16647 | 7.300±0.084 | 6.810 ±0.074 |
| Icelandic-IcePaHC | 44039 | 6.286±0.072 | 6.330 ±0.060 |
| Japanese-GSD | 57109 | 7.676±0.042 | 7.238 ±0.105 |
| Korean-Kaist | 27363 | 9.585±0.127 | 6.106 ±0.094 |
| Latvian-LVTB | 15984 | 5.739±0.080 | 3.584±0.092 |
| Norwegian-Bokmaal | 20044 | 5.101±0.054 | 3.245±0.100 |
| Persian-PerDT | 29107 | 7.038±0.072 | 5.330±0.070 |
| Polish-PDB | 22152 | 5.668±0.091 | 3.962 ±0.084 |
| Romanian-Nonstd | 26225 | 5.212±0.061 | 4.298±0.082 |
| Russian-SynTag | 87336 | 5.696±0.094 | 3.821±0.063 |
| Spanish-Ancora | 17662 | 5.243±0.070 | 5.195±0.079 |
| Turkish-Kenet | 18687 | 8.647±0.052 | 6.123±0.068 |
| English-EWT re-parsed by parsers | | | |
| English-EWT-Spacy | 16621 | 9.479±0.121 | 5.168 ±0.090 |
| English-EWT-UDPipe | 16621 | 4.178±0.100 | 2.630 ±0.126 |
| English-EWT-ChatGPT4o | 16621 | 6.111±0.110 | 3.165±0.127 |
| English-EWT-ChatGPT5 | 16621 | 4.634±0.071 | 2.560±0.076 |
| Random Trees | | | |
| Rand-Star | 100000 | 12.264±0.129 | 8.640±0.138 |
| Rand-Balanced | 100000 | 12.617±0.079 | 9.411±0.113 |
| Rand-Uniform | 100000 | 9.390±0.094 | 9.068±0.092 |
| Rand-Linear (ord=2) | 100000 | 16.581±0.208 | 10.270±0.061 |
| Rand-Markov (o=3) | 100000 | 13.042±0.134 | 10.466±0.079 |
| Rand-Markov (o=5) | 100000 | 11.324±0.103 | 10.353±0.135 |
| Rand-Markov (o=10) | 100000 | 10.167±0.137 | 10.026±0.107 |
| Rand-English-EWT-CF | 100000 | 6.554±0.073 | 9.701±0.142 |
| Rand-GPT2 | 12880 | 5.116±0.065 | 4.280±0.095 |
| Rand-GPT3 | 14327 | 5.169±0.039 | 3.884±0.101 |
| Rand-GPT4o | 19327 | 5.263±0.062 | 4.060±0.081 |
| Rand-GPT5 | 15996 | 5.464±0.065 | 3.226±0.097 |

in the first block, 100,000 random context-free trees were generated by building a grammar from each corpus. The second block of "Random Trees" simply lists Rand-English-EWT-CF, built from English-EWT as an example.

**ChatGPT Trees** appear in the last block. ChatGPT rephrases/translates a given text or generates an answer for a given question. Because rephrasing/translating produces texts that are constrained to the original input sentence structure, we generated texts by prompting ChatGPT with questions.

We first downloaded questions from SQuAD (Rajpurkar et al., 2016), a dataset for question answering in the field of NLP. The questions (without answers) were fed to random ChatGPT versions via the OpenAI API, yielding one to multiple answer sentences per question. Among a substantial number of LLMs, here we focus on ChatGPT 2, 3, 4o, and 5 as representatives. Our dataset includes over 10,000 sentences for each model, listed as Rand-GPT{2,3,4o,5} in the last block of Table 2.

### 4.3 EXPERIMENTAL SETTINGS AND BASELINES

To train the function $f$ in $h = f(t)$ (section 3.2), we used AdamW (learning rate $3\times10^{-4}$, weight decay $10^{-4}$), cosine annealing over 10 epochs, a batch size of 32, and gradient clipping at 1.0. The implementation was in PyTorch with a GPU.

The models acquired by training different sets of language classes were experimentally stable topologically, but the distance value ranges could be different. We considered two models: a **common model** trained on 21 natural languages (first block, Table 2), and an **individual model** trained on a particular set of datasets. Because our model interpolates/extrapolates the untrained trees, the two models' results were similar. Therefore, in the main text, we present the results acquired with the individual model, all trained on the corpora under examination. Supplementary M gives all the corresponding results using the common model for comparison. The last column of Table 2 lists the Wasserstein distances for English-EWT obtained from the *common model*, so the values are comparable vertically.

The model's two hyperparameters were decided as follows. First, $H$ should be as small as possible. Supplementary G shows how the performance is similar for $H \geq 128$, and we thus used $H = 128$ in this work. Second, $\alpha$ represents a tradeoff between tree-structure and class prediction. From the analysis given in Supplementary H, we used $\alpha = 0.8$.

We sampled trees by non-replacement. *Per class*, for training we sampled $K = 10,000$ trees, and for validation we took $S = 1,000$ out-of-sample trees. These choices of $K, S$ were guided by our convergence studies reported in Supplementary E and F, respectively. When we needed the average and standard deviation, we used $I = 10$ trials of $S$ set samples, where a sample could be included across different trial sets.

For visualization we used t-SNE (Van der Maaten & Hinton, 2008) on the pairwise distance matrix $D_{ij} = m(h_i, h_j)$ between embedded trees. For clarity, we adjusted and trimmed the t-SNE settings, as given in Supplementary I. Class centroids in the 2D maps are shown by large "X" markers.

Our baselines comprised the tree metric and prediction models in Table 1. The former included TED, tree kernels, and PQ-grams. TED is the most fundamental, with every edit cost set to 1. The details of tree kernels and PQ-grams are given in Supplementary K. For the latter, we used TreeLSTM (Tai et al., 2015) and TreeTransformer (Wang et al., 2019) because both families entail general-purpose encoders, as explained in Supplementary J.

Note that, in the following experimental sections, the same experimental conditions were applied for comparison among the baselines, in terms of the (i) inputs/preprocessing, (ii) supervision, (iii) model size (matched hidden size, depth), (iv) optimization/data (same optimizer, schedule, batch size, epochs, seed, splits), and (v) evaluation (Wasserstein distance applied to the ground metric, Euclid for Tree-LSTM/Transformer). When obtaining TED values, however, because it is too slow, we used $S = 200$ samples instead of $S = 1000$ (except in section 5.1).

## 5 EVALUATION

### 5.1 SPEED

Our method has a great advantage in its computational complexity for distance calculation with respect to the tree size, as seen in the fourth column of Table 1. Speed evaluation only applies to the first block's methods because the prediction models in the second block don't represent trees. Table 3 lists the random-pair distance calculation times for $S = 1000$ pairs of random trees from ENGLISH-EWT and $I = 10$ trials.

Table 3: Mean speed to calculate $S = 1000$ tree-pair distances (second).

| Method | mean |
| --- | --- |
| TED (ZSS) | 23.2193 |
| Tree kernels | 2.8227 |
| PQ-grams | 0.0141 |
| Ours | **0.0075** |

Our method is 3000 times faster than TED, 400 times than tree kernels, reflecting its lower computational complexity. Note that our model is fast in training, too: a machine with eight RTX4090 GPUs took only a few hours to build the common model across 210k ($=21 \times 10$k) trees. PQ-grams are also fast, but our method is twice as fast and provides better quality, as will be seen in section 5.3.

### 5.2 PREDICTION TASKS

Our model is evaluated by predicting the tree structure and class (language). Regarding the baselines, because the previous tree metric methods do not apply without a nontrivial prediction module, we compare the performance with the prediction models, i.e., Tree-LSTM/Transformer.

Specifically, we compare the accuracy of the token position and parent index prediction within the sentence, and language class. We used the following three settings:

**S1: In-language.** The model learns two samples from the same set of English-EWT.

**S2: 5 Languages,** consisting of English-EWT with four other treebanks Japanese-GSD, Korean-Kaist, Icelandic-IcePaHC, and Hindi-HDTB, the four farthest from EWT.

**S3: All UD treebank natural languages in Table 2.** The model trains for English-EWT and the 20 other languages in the first block of Table 2.

The accuracy ranges between 0 and 1. The value is better when closer to 1, *except* for S1-language, which should be 50%.

Table 4 lists the mean accuracies over $I = 10$ trials (with small standard deviations, around 0.03), for every model in the second column trained on the scenarios in the first column. For the token position (third column), Tree-Transformer was better in the *few-class* scenarios of **S1** and **S2**, as Tree Transformer is given an attention prior specially designed to

Table 4: Prediction task accuracy (for S1-language the best is 50%, i.e., a 50-50 chance; otherwise, the larger the better).

| Scenario | Model | position | parent | language |
|---|---|---|---|---|
| **S1**:EWT A vs. EWT B | TrLSTM | 12.68% | 32.84% | 48.16% |
| | TrTransformer | **47.81%** | 41.51% | 51.25% |
| | Ours | 45.76% | **53.98%** | **49.92%** |
| **S2**:EWT EWT and Far 4 | TrLSTM | 11.68% | 28.72% | 46.20% |
| | TrTransformer | **51.72%** | 39.88% | 73.71% |
| | Ours | 49.16% | **55.06%** | **89.17%** |
| **S3**:All UD in NL Table 2 | TrLSTM | 11.69% | 26.89% | 12.70% |
| | TrTransformer | 54.37% | 64.68% | 41.27% |
| | Ours | **66.88%** | **67.66%** | **43.32%** |

model/predict the intra-sentence structure. In the *many-class* scenario **S3**, however, TreeTransformer generalized ineffectively across typologically diverse structures, where our model outperformed it dramatically. Our model also consistently outperformed the prediction models on the parent index (fourth column) and language prediction (last).

### 5.3 Quality of Tree Metric Space

Among tree-metric methods, TED has been widely applied despite its computational complexity because it provides a reasonable tree distance that matches engineering needs well. This also applies here for sentences. The third column of Table 2 lists the Wasserstein distances of language corpora from English-EWT for TED. The relative values are intuitively comprehensible: the first row, with two English-EWT samples, has an average distance of 2.98, while all other distances are larger, with Indo-European languages closer to English.

We evaluated the correlation with the TED distances listed for all classes in Table 2, i.e., the Pearson correlation between the third and fourth columns for ours, and the corresponding evaluations for tree kernels and PQ-grams. Note that only ours supervises the

Table 5: Correlation with TED.

| Kernel | PQ-gram | Ours |
|---|---|---|
| 0.29 | 0.79 | 0.85 |

language classes, whereas TED, tree kernels, and PQ-grams don't. Despite this difference, the metric acquired with ours correlated the best with TED, as seen in Table 5, suggesting how our method learns a TED-like distance even without TED supervision. PQ-grams also showed surprisingly good correspondence, which is reported here for the first time, and it is a fast method, while the kernel methods produced lower correlation. They could correlate better if they were trained on tree sentences, but such work is nontrivial and beyond our scope here.

To sum up, our model produces a TED-like tree metric for sentences that *doesn't* use TED supervision, is low in complexity, and is capable of tree structure analysis and class prediction.

## 6 Applications of Our Tree Representation

### 6.1 Evaluating Parsers and ChatGPT-Generated Trees

Our model can evaluate recent techniques in a new way. The first such application is parser evaluation, which is necessary to conduct further analyses with ChatGPT tree evolution.

The left graph in Fig. 3 shows English-EWT (red), and its sentences as reparsed by Spacy (green) and UDPipe (blue) (Straka, 2018; Straka & Straková, 2020). It also shows the tree acquired with ChatGPT5 (black) by prompting English-EWT sentences to be transformed to the UD structure.

The reparsed UD structures should be mixed with English-EWT when being parsed by an ideal parser. This is the case for UDPipe because it is trained on UD data, whereas we see a large discrepancy for Spacy. On the other hand, ChatGPT5 shares the same region with English-EWT. The third

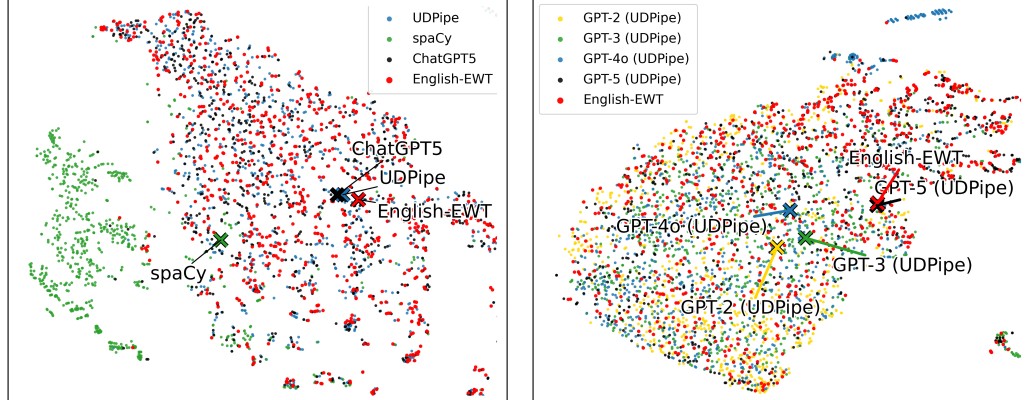

Figure 3: Left: English-EWT (red) and its reparses by UDPipe (blue), Spacy (green), and ChatGPT5 (black). Right: English-EWT and texts generated by ChatGPT {2 (yellow), 3(green), 4o(blue), 5(black)}, reparsed by UDPipe.

and fourth columns of second block of Table 2 give the Wasserstein distances from English-EWT for TED and ours, respectively. GPT5 is way closer to English-EWT than GPT4o, even closer than UDPipe with our method. The accuracies of reparsed English-EWT were 87.36%, 21.23%, 51.83%, and 76.21% for UDPipe, Spacy, ChatGPT4o, and 5, respectively, agreeing with the figure and the distance. As our work uses UD as its basis, we used UDPipe in subsequent work with ChatGPT.

We parsed the texts generated by ChatGPT 2-5o, as mentioned in section 4.2, with UDPipe, and the right graph in Fig. 3 shows the resulting metric space. Whereas all trees are from English-EWT sentences in the left graph, the sentences here are all different. We can see the concentration of ChatGPT models, especially 2 and 3, farther from English-EWT. The mean Wasserstein distances from English-EWT, given in Table 2 (last column), show how GPT approached closer to English-EWT through the generations from 2 to 5. This trend was not captured by TED (third column).

## 6.2 RANDOM TREES VS. NATURAL LANGUAGE

To improve language technology, including language models, a basic understanding of the characteristics of natural language sentence structure would be beneficial. Hence, this section considers natural language among various random trees.

In Fig. 4, the left graph shows a t-SNE depiction of all the natural languages and random trees, i.e., the first and third blocks of Table 2. We first see that simple random trees appear entirely separate from natural language trees, toward the top. As for the context-free (CF) case (in blue), the points are closer to NL (in red), but their respective points appear in different areas. Note that when the two classes are the same, our model correctly predicts at a near-chance level of 50% (see Table 4, top right cell). This result suggests the limitation of context-free modeling of natural language. Supplementary N gives partial results for English and English-CF, which are pretty separated.

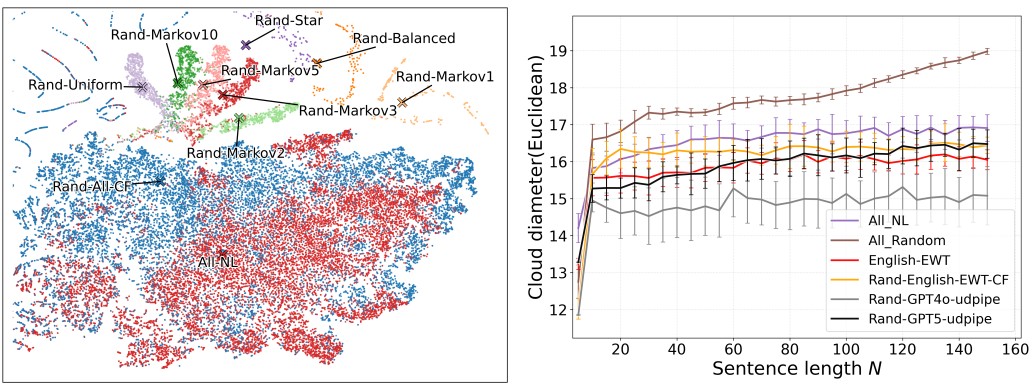

Figure 4: Left: Space of all natural languages (ALL-NL, red) in Table 2, all context-free trees (Rand-ALL-CF, blue), and other, simpler random trees. Right: Diameters of the tree sets of sentence samples of length $\leq N$.

The right graph in Fig. 4 shows the mean diameter $D$ of the point clouds—the distance between the two farthest points in a cloud—for sentences of length $\leq N$. Because the distance must be comparable, we used the common model trained only on all natural language trees (whereas the left graph included random trees for training too). The curves show the diameter growing rapidly with the sentence length up to less than $N = 20$, with a more gradual increase for longer sentences. The change around 20 appears because we sampled lengths of random sentences from a real sentence-length distribution (see Fig. 6, Supplementary C). The vertical order of the curves is reasonable, with the diameter for ALL_Random being larger and increasing faster than for All_NL and the English dataset. While $D$ was small for the ChatGPT4o sentences, suggesting limited sentence structure, the ChatGPT5 sentence diameter was closer to English-EWT, showing the great improvement of ChatGPT5.

For a TED metric space of cost 1, the diameter among trees of length $n$ is obviously $2(n - 1)$ (removal and addition of a star tree to produce a linear tree for all nodes except the root). This figure suggests that the obtained space with our method compares to a fish-eye lens of a TED space, highlighting the most important trees of shorter sentences.

### 6.3 NATURAL LANGUAGE TREES AND UD CORPORA EVALUATION

Figure 5 shows point clouds of all the natural language datasets. The clustering reflects language families, with clusters of Romance/Slavic languages, SVO languages in the middle, and SOV languages toward the bottom. Despite our method not using any words or POSs, the figure reveals these linguistic characteristics. The analysis also applies to the intra-language case (for English, Supplementary L).

Usefully, this result provides new evidence for known problems of UD corpora. For example, while Japanese and Korean should be better mixed from a linguistic perspective, their point clouds appear separately at the bottom. (Han et al., 2020) reported UD annotation issues for Japanese and Korean, suggesting the need to revise the UD framework itself. Further analysis with our method could provide insights for improvement in UD itself and phylogenetic reconsideration of languages.

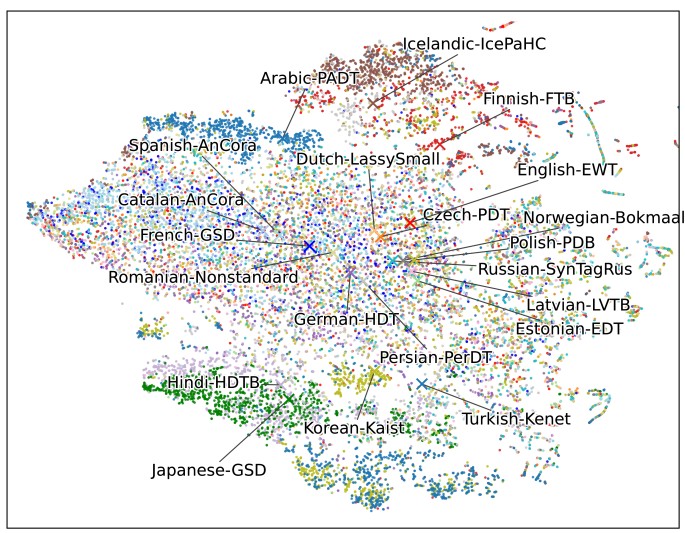

Figure 5: All trees sampled with $S = 1000$ per natural language, and their centroids in the acquired tree metric space.

## 7 CONCLUSION

We presented a sentence tree metric space built by representation learning of embedding trees in a vector space. The proposed method has lower computational complexity than previous tree metric methods, including the tree edit distance (TED), tree kernels, and PQ-grams, enabling large-scale comparison among trees. On prediction tasks, our method mostly outperforms the previous, representative tree prediction methods, tree-LSTMs/Transformers. Furthermore, the resulting metric correlates well with TED among sentence trees, without requiring TED values for supervision.

We showed the effectiveness of our metric space in three ways. First, it enables evaluation of parsers and LLMs in terms of how well they capture natural language tree structure. Second, it enables analyses of natural language with respect to random trees, in particular highlighting the limitation of modeling natural language as being context free. Lastly, it provides good means of assessing intra- and inter-linguistic relations, thus providing evidence for future corpora revisions.

**Reproducibility Statement (not counted within the page limit):** All code associated with this work will be made publicly available upon the publication of this paper. The parameters used are fully documented within the paper. This work is based on publicly available UD data. In addition, the random trees generated and used in this study will also be released publicly at the time of publication.

**Code of Ethics:** This paper contains no content to which a code of ethics is applicable.

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

# Supplementary Material

## A WASSERSTEIN DISTANCE BETWEEN TWO SETS OF TREES

Given a tree metric space $(T, m)$ and two finite sets of trees $T_1, T_2 \subset T$, there are several measures to compute the degree of difference between them. Here, we use the $p$-Wasserstein metric on the empirical distributions:

$$m_{W_p}(T_1, T_2) := W_p \left( \frac{1}{n_1} \sum_{i=1}^{n_1} \delta(t_i), \frac{1}{n_2} \sum_{j=1}^{n_2} \delta(t_j') \right), \tag{1}$$

where $T_1 = \{t_i\}_{i=1}^{n_1}$, $T_2 = \{t_j'\}_{j=1}^{n_2} \subset T$, and $\delta(t)$ is the Dirac measure at $t \in T$. Here, $\frac{1}{n_1} \sum_{i=1}^{n_1} \delta(t_i)$ and $\frac{1}{n_2} \sum_{j=1}^{n_2} \delta(t_j)$ become probability measures. The $p$th Wasserstein metric between two probability measures $\mu, \nu$ on $T$ with a finite $p$th moment is defined as

$$W_p(\mu, \nu) := \left( \inf_{\gamma \in \Gamma(\mu, \nu)} \int_{T \times T} m(t, t')^p \, \mathrm{d}\gamma(t, t') \right)^{1/p}, \tag{2}$$

where $\Gamma(\mu, \nu)$ is the set of all couplings of $\mu$ and $\nu$, i.e., measures on $T \times T$ with marginals $\mu$ and $\nu$ on the first and second factors, respectively. The Wasserstein metric can be recognized as a generalization of the optimal transport cost. See, for example, (Villani, 2008) and (Peyré & Cuturi, 2019) for more details on the theory, computation, and application of the Wasserstein metric.

The motivations of using the Wasserstein metric for our tree metric space are primarily that it can use the metric function between points and the Wasserstein metric is itself a metric.

Simple other possibilities include geometric medians, or the Hausdorff distance, which are inadequate here, only capturing the set distance through two particular points. The typical other options include the centroid distance, but here we want to use the distance based on the distribution of points.

If the problem is formulated as the distance between probability measures, the Kullback-Leibler divergence or the total variation distance might be candidate metrics. However, both of those use almost no information of the ground metric space. For example, if there is a tree $t$ such that $t \in T_1$ but $t \notin T_2$, then neither measure's value changes no matter how far $t$ is from the elements of $T_2$.

Alternatively, we could consider the distance between $T_1$ and $T_2$ as a dissimilarity of clusters. One of the most popular criteria for clustering analysis is the group average:

$$\frac{1}{|T_1||T_2|} \sum_{t \in T_1, t' \in T_2} m(t, t').$$

However, this is not a distance because it fails to be 0 even when $T_1 = T_2$; thus, it may overestimate the distance between clusters with large variances.

Among all options, the Wasserstein distance best suits a metric based on the point distribution, given its reasonable computational complexity (Sinkhorn & Knopp, 1967) and robustness with outliers.

## B TREE NODE FEATURES

Before being fed to the Transformer, each node is represented as a vector of $A$ features, and trees are linearized in depth-first order with a validity mask for padding. We use the following six features identically across all models:

- **DFS Node position:** integer in $\{0, \ldots, n-1\}$ indicating the node's DFS order.
- **Node position:** integer in $\{0, \ldots, n-1\}$ indicating the index in the sentence.
- **Parent position:** integer in $\{0, \ldots, n-1\}$ indicating the parent's DFS position in the sentence.
- **Depth:** integer depth of the node from the root.

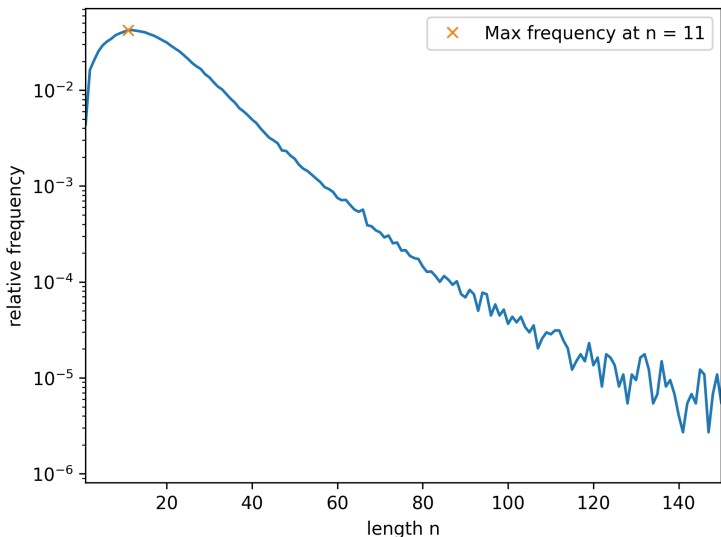

Figure 6: Distribution of sentence lengths for all natural language data we used.

- **Modifying direction:** categorical feature encoding the attachment direction relative to the parent, with -1 before and +1 after.
- **Arity**: number of node's children.
- **Number of Siblings:** number of siblings sharing the same parent.
- **Position among Siblings:** order among siblings.
- **Sentence size:** $n$, the tree's length (number of nodes).
- **Position of Sentence Head :** integer in $\{0, \ldots, n-1\}$ indicating the sentence's head position.

## C  SIMPLE RANDOM TREES

For simple random trees, the size $n$ could be infinite. As we want to focus on natural language sentences, we first sampled the tree length $n$ from the sentence length distribution of natural language UD trees. Figure 6 shows the distribution of all UD sentences we used.

For a given $n$, a tree is constructed recursively by inserting the $j$th node in a tree of size $j-1$, $j \leq n$. The order of children is defined by the order of their creation.

We considered the following variations in this paper:

**Star:** All nodes share a common root node with all other nodes as its children.

**Balanced:** The complete binary tree of $n$ nodes (levels filled left to right).

**Uniform:** A new node is attached uniformly randomly to one of the tree's previously generated nodes.

**Linear:** Each node except the leaf node always has a single child. This forms a Markov tree of order 2.

**Markov:** For $j > 0$, a new node is attached to one of the $j$ previous nodes in the tree. The case of $j = 1$ is a linear tree (of order 2), as mentioned above, while the cases of $j = 2, 4, 9$ are examined in this work (Markov orders $3, 5, 10$, respectively).

After a tree is sampled, a signed ordered tree is sampled. For each parent $\rightarrow$ child edge, we attach a direction sign: "+1" if the child appears to the right of its parent in linear order, and "-1" if it appears to the left. The root has no sign. For Markov, the children's sign is always +1 by definition.

On the other hand, children of the first three kinds can have "-1" and "+1". Since natural language trees are non-projective (dependency branches do not cross any other), we sample only projective trees: for a node with $b$ children, we sample an integer $c \in [0, b)$. As the children are already ordered when generated, $[0,c]$ has "-1" while the rest have "+1".

## D    RANDOM CONTEXT-FREE TREE GENERATION

Our set of context-free random trees was generated from an UD corpus, as follows. First, a dependency grammar was constructed for the corpus by examining each word $w$ and recording what other words modify it. As $w$ can be modified by words before or after itself, and because we are using a signed ordered tree, the sets of modifying words *before* and *after* $w$ are recorded with their frequency counts, together with the distribution of the numbers of modifiers. The resulting sets of modifier-modified relations before and after $w$ are denoted as $Gb$ and $Ga$, respectively. Furthermore, all main predicates are collected from the corpus together with their frequencies; the resulting set is denoted as $Gh$.

From this grammar, trees are generatively sampled. A main predicate is sampled from $Gh$ in proportion to its frequency. Then, a random sentence is generated recursively by using a function $F$, starting with the main predicate $w$ as the target word. For every target word $w$, from each of its $Gb$ and $Ga$, $F$ first samples the number of modifiers; then, it samples the corresponding number of modifiers in proportion to the frequency of the number of occurrences. For each modifier generated in this way as a new target, the function $F$ is called recursively. This recursive procedure stops when the number of modifiers is zero.

In the grammar construction, $Gb$ and $Ga$ are empty for many words that occurred in the tree without a modifier. The recursive procedure thus terminates when a zero is sampled as the number of modifiers. In other words, our sample procedure terminates without introducing any arbitrary setting such as a stop probability.

## E    CONVERGENCE OF PROPOSED MODEL WITH RESPECT TO SAMPLE SIZE $K$

We study how the learned tree metric stabilizes as the *training sample size* $K$ increases, while keeping all other settings fixed following the main text (inputs, model, optimizer, and hyperparameters $S = 1000$ and $I = 10$ used for distance estimation).

For each $K$, we train the model from scratch, embed out-of-sample trees, and compute the Wasserstein (Sinkhorn) distance between sets of tree embeddings.

Two scenarios are considered:

- **Same-set stability (EWT↔EWT), s1** of section 5.2. Both sets are sampled from English-EWT: this confirms whether our metric stabilizes to a certain value when the underlying distributions are the same.

- **Multi-set stability (EWT↔Others), s2** of section 5.2. One set is English-EWT and the other is drawn from Japanese-GSD, Korean-Kaist, Icelandic-IcePaHC, and Hindi-HDTB. For each $K$, we compute EWT↔JA/IC/KO/HI distances to summarize the cross-dataset behavior.

**Training sizes.**    For both scenarios we sweep $K \in 500, 1500 \cup 2000, 3000, \ldots, 12000$.

**Findings.**    (*i*) **Same-set.** The Wasserstein distance increases rapidly as $K$ grows and forms a clear plateau around $\mathbf{K} \approx \mathbf{9{,}000}$, with small standard deviations thereafter. (*ii*) **Multi-set.** When $K < 8{,}000$, both the mean distance and its standard deviation fluctuate noticeably; once $K \geq 10{,}000$, the distances and error bars shrink and stabilize. Considering the total corpus size and training budget, we adopt $\mathbf{K} = \mathbf{10{,}000}$ in all main experiments as a conservative, stable operating point.

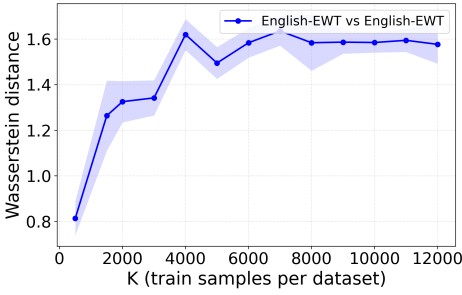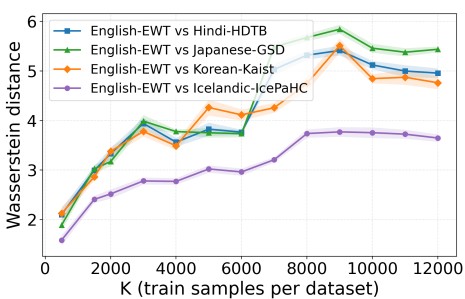

Figure 7: **Convergence w.r.t. training size $K$. Left:** EWT vs. EWT (same-set). The distance rises quickly at small $K$ and plateaus near $K \approx 8000$; the error bars are mean±std over $I = 10$ trials. **Right:** EWT vs. each of Hindi-HDTB, Japanese-GSD, Korean-Kaist, Icelandic-IcePaHC (multi-set), showing three non-monotonic curves with larger variability for $K{<}8000$ and stabilization for $K{\geq}10{,}000$. We thus adopt $K{=}10{,}000$ in the main text.

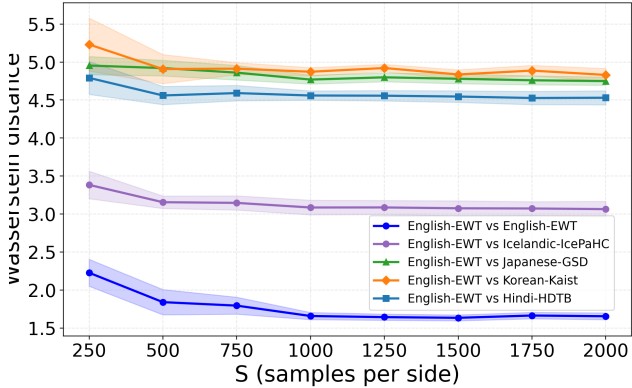

Figure 8: **Convergence w.r.t. sample size $S$.** The distance fluctuates at small $S$ and becomes stable with a low std after $S{\approx}1000$; the error bars are mean±std over $I = 10$ trials.

## F  SAMPLE SIZE $S$

We considered the sample size $S$ via the stability of the Wasserstein distance. We jointly trained on five languages (English-EWT, Hindi-HDTB, Japanese-GSD, Korean-Kaist, Icelandic-IcePaHC, **s2** of section 5.2) and swept $S \in \{250, 500, ..., 2000\}$, keeping all the other settings identical to our model settings in the evaluation. The distance became stable and the std became small when $S \geq 1000$. Considering the data size and computational cost, we use $S = 1000$ as a conservative, stable operating point.

## G  REPRESENTATION DIMENSION $H$

We trained on **S2**: 5-languages of section 5.2, with English-EWT and four other treebanks (Japanese-GSD, Korean-Kaist, Icelandic-IcePaHC, and Hindi-HDTB, the four farthest languages from EWT), and we swept $H \in \{64, 96, 128, 192, 256\}$ (fixing $\alpha = 0.8$). We report the mean±std accuracy on three tasks: node position (ID), parent position (PID), and language (LANG). The results are shown on the left in Fig. 9.

**Findings.** The accuracy improved sharply from $H{=}64$ to $H{=}128$ on all tasks, then *plateaued* for $H \geq 128$ with only minor, non-systematic fluctuations (small std across runs). Smaller $H$ underfits (notably on Position-ID), while larger $H$ presented tendency to overfit.

**Decision.** We adopt $\boldsymbol{H = 128}$ as the smallest near-peak configuration that balances accuracy and efficiency.

# H   HYPERPARAMETER $\alpha$

Using the same language setup as to verify $H$, we swept $\alpha \in \{0.0, 0.1, 0.3, 0.5, 0.6, 0.8, 1.0\}$ (fixing $H = 128$) and evaluated the node position (Seq-ID), parent position (Parent-ID), and language (Lang-ID). The results are shown on the right in Fig. 9.

**Findings.** All three tasks reach their *jointly highest* region near $\boldsymbol{\alpha = 0.8}$. Extremes harm balance: low $\alpha$ underweights the language signal (hurting Lang-ID), while high $\alpha$ overweights it (hurting structure tasks). Around $\alpha = 0.8$, the three curves are simultaneously high and stable (small std).

**Decision.** We adopt $\boldsymbol{\alpha = 0.8}$ as a robust operating point that maintains strong structure prediction while achieving high language discrimination, consistent with our paper's emphasis on modeling languages in a unified tree metric space.

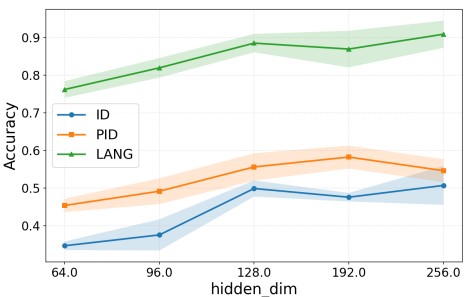 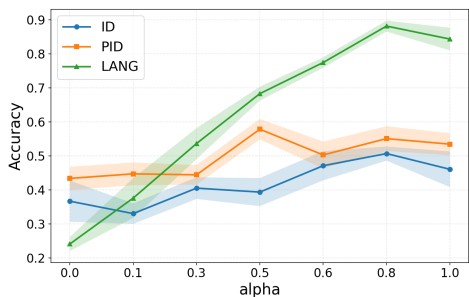

Figure 9: **Left:** Accuracy vs. representation dimension $H$; performance rises steeply up to $H = 128$ and then plateaus or drops (fluctuates). **Right:** Accuracy vs. $\alpha$; all three tasks peak jointly near $\alpha = 0.8$. Both panels were trained jointly on English-EWT, Japanese-GSD, Korean-Kaist, Icelandic-IcePaHC, and Hindi-HDTB; the curves show node position (ID), parent position (PID), and language class (LANG).

# I   T-SNE SETTINGS

In our t-SNE figure, outliers appear scarcely scattered outside the main point clouds. To achieve clarity and limit the space taken only by outliers, we filtered out a small number of outliers and trimmed the figure as follows.

Before running t-SNE, we applied per-class kNN sparsity trimming in $h$-space: after $z$-score normalization, each point was scored by its mean $k$-nearest-neighbor distance ($k=10$), and the top 1% highest-score points were removed to suppress outliers while preserving the cluster shape. t-SNE was then run on the metric distances $D_{ij} = m(h_i, h_j)$ among the retained points.

The obtained figure still has some space for outliers, and we trimmed the figure edges to suppress the amount of white space taken by them.

# J   PREDICTION MODEL BASELINES

## J.1   TREELSTM (TAI ET AL., 2015)

This section summarizes two prediction model baselines used in the evaluation.

**What it is.** The Child-Sum TreeLSTM (Tai et al., 2015) extends a sequential LSTM from a chain to a tree: each parent state is composed from its children. For a node $j$ with children $C(j)$, it aggregates child hidden states,

$$\tilde{h}_j = \sum_{k \in C(j)} h_k,$$

then applies LSTM gating conditioned on $(x_j, \tilde{h}_j)$, including a child-specific forget gate $f_{jk}$ for each $k \in C(j)$. If every node has exactly one child, it collapses to a vanilla sequential LSTM.

**How it differs from vanilla LSTM.** A standard LSTM propagates along a single time chain; TreeL-STM performs bottom-up composition over the *tree*, allowing information to flow from multiple children into a parent at each step.

**Application to our evaluation of section 5.2**

**Inputs.** For fairness, each node $j$ uses the same feature bundle as our main model, the hidden dimension matches the main model ($H = 128$).

**Architecture.** We use the Child-Sum TreeLSTM (Tai et al., 2015) bottom-up. The sentence representation is the mean of node states (same as our main model), $h_{\text{sent}} = \frac{1}{N} \sum_{j=1}^{N} h_j$.

**Training.** Identical to our model, we train using the loss function of cross-entropies using the linear heads of node/parent positions $\mathcal{L}_{\text{predict}}$ and class $\mathcal{L}_{\text{class}}$, with a total loss of $\mathcal{L} = \mathcal{L}_{\text{predict}} + \alpha \mathcal{L}_{\text{lang}}$, and $\alpha$ shared with the main model. Other settings are all identical to our model training process.

### J.2 Baseline II: Tree Transformer (Yang & Dredze, 2016; Wang et al., 2019)

**What it is.** A Tree Transformer injects *tree bias* into self-attention so that tokens within the same constituent/subtree attend more strongly. At each layer, attention is modulated as $E$ by a structural prior $C \in [0, 1]^{n \times n}$:

$$E = C \odot \text{softmax}(QK^\top / \sqrt{d}),$$

where $C_{ij}$ encodes the prior affinity that positions $i$ and $j$ belong to the same constituent, and $Q, K, d$ are the standard Transformer Query, Key, and their dimension, respectively. When $C$ is all-ones, this reduces to a vanilla Transformer.

**How it differs from vanilla Transformer.** A vanilla Transformer uses content-only attention, whereas the Tree Transformer multiplies attention by $C$ to favor within-subtree pairs, injecting a structural inductive bias while preserving long-range interactions.

**Application to our evaluation of section 5.2.** For fairness, we use the Tree Transformer as a replacement for our model while keeping everything else identical (inputs, set of features, masking scheme, heads, objectives, optimizer/schedule, training steps).

## K Baselines of Tree Metrics

We have three tree metric space baselines, and our setting is summarized as follows:

**TED:** We use the most basic static version with all edit costs as 1 and the ZSS library.

**Tree Kernels:** We use the *exact* Collins–Duffy Subset Tree Kernel (SST) with decay $\lambda$ (we set $\lambda = 0.4$). Kernel values $K(x, y)$ are converted to distances via the kernel-induced metric:

$$m_{\text{TK}}(x, y) = \sqrt{\max\{0, K(x, x) + K(y, y) - 2K(x, y)\}},$$

**PQ-Grams:** We use the library PyGram, and $p = 2$ and $q = 3$.

Table 6: Three English datasets with #sentences and mean Wasserstein distances from English-EWT for TED and Ours , through $I = 10$ trials.

| treebank | # trees | TED | Ours |
|---|---|---|---|
| English-EWT | 16621 | 2.984±0.173 | 2.417±0.080 |
| English-Atis | 5432 | 7.272±0.048 | 6.461± 0.123 |
| English-ESL | 5124 | 5.057±0.063 | 4.263±0.063 |

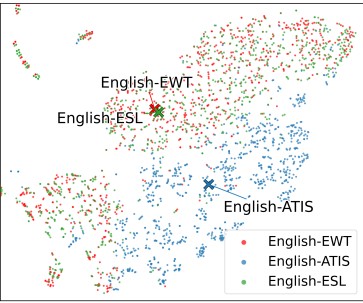

Figure 10: Three point clouds of English datasets using the common model built only from 21 natural languages: English-EWT (written), English-Atis (speech), and English-ESL (written by non-natives).

## L    COMPARISON AMONG DIFFERENT DATASETS IN ENGLISH

Table 6 shows three English datasets, for language written by natives (EWT), and for language spoken (Atis) and written by non-natives (ESL). Figure 10 shows the space obtained from the common model, trained on 21 natural languages, and hence the model doesn't have information on the language classes of Atis and ESL. Spoken languages appear clearly distant from written, whereas ESL stays mixed among EWT. This figure confirms our intuitive understandings, while quantifying the actual distances among classes.

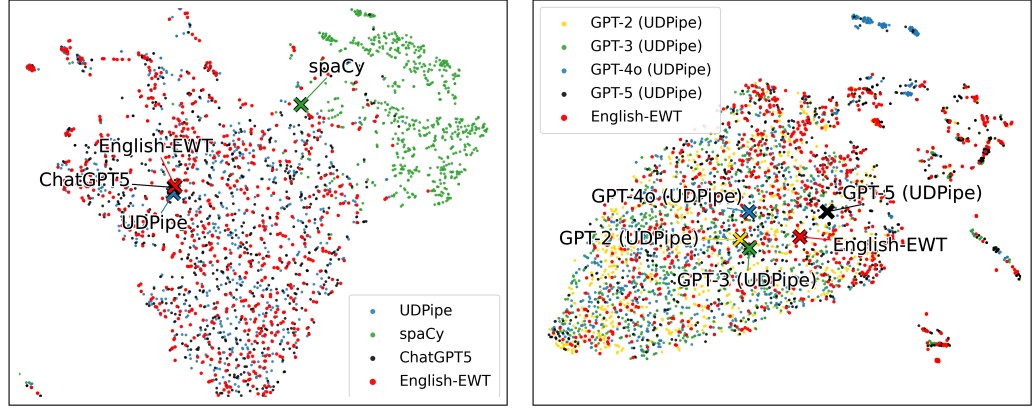

Figure 11: Figure 3 but with the common model only trained for 21 natural languages. Left: English-EWT (red) and its reparses by UDPipe (blue), Spacy (green), and ChatGPT5 (black). Right: English-EWT and texts generated by ChatGPT 2o (yellow), 3o (green), 4o (blue), and 5o (black), parsed by UDPipe.

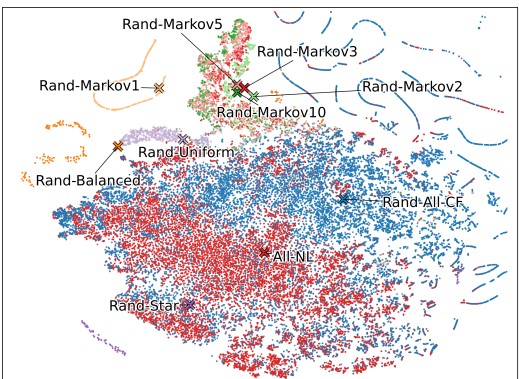

Figure 12: Fig. 4, left, but using the common model: the space of all natural languages (ALL-NL, red) in Table 2, all context-free trees (Rand-ALL-CF, blue), and other, simpler random trees.

## M  FIGURES 3 AND 4 FOR *Common Model* TRAINED ONLY ON NATURAL LANGUAGE

This section shows additional versions of the results in Figs. 3 and 4 (left) when using the **common model**, trained on 21 natural languages (see section 4.3). The main understandings remain the same, showing that the model can interpolate/extrapolate unseen datasets, and that it is stable across datasets.

In Fig. 11, the figures are differently transposed by t-SNE from Fig.3. Although the presentation among points is little different, the relative understanding remains the same, as discussed in section 6.1.

As for Fig. 12, because the model was not trained with random trees, the model deduced their locations by *interpolating/extrapolating* natural language trees. Unlike in the left graph of Fig. 4, the simple random trees are more aggregated in a peninsula at the top. Despite this model having no context-free information, it extrapolated similarly to the figure presented in the main text. The basic understanding gained from this figure is the same as that discussed in section 6.2.

## N  ENGLISH-EWT AND ITS CONTEXT FREE TREES

Figure 13 shows English-EWT and Rand-English-EWT-CF, its context-free version. The left graph was generated by a model solely trained on these two datasets, whereas the right graph is for the common model, trained on 21 languages, with partial plots of Fig. 12 only for English-EWT and its context-free version. While the placements of the plots are rotated and different, they appear separately for both.

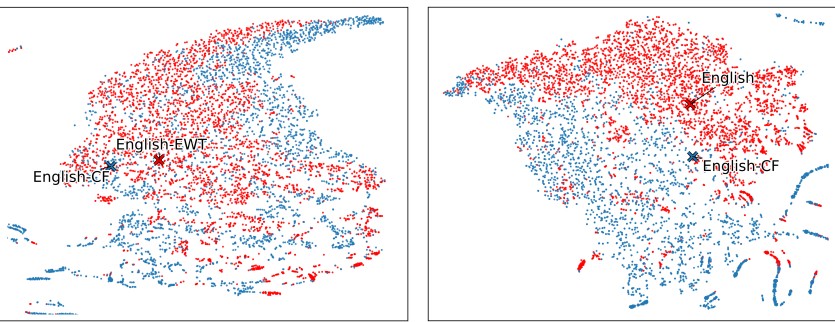

Figure 13: English-EWT (red) and its context-free version (blue). The left graph is for the model trained solely on English-EWT and Rand-English-EWT-CF, whereas the right graph is for the common model trained on 21 natural language corpora.