# OpenReview forum: "Representing Sentence Structure in a Tree Metric Space"
_ICLR.cc/2026/Conference — ICLR 2026 Conference Withdrawn Submission_

### Official Review · Reviewer_PzR7 · 2025-10-30

**Soundness:** 3
**Presentation:** 3
**Contribution:** 2
**Rating:** 6
**Confidence:** 2

**Summary:**

This paper introduces a new framework for representing sentence structures in a tree metric space via representation learning. Each dependency tree is embedded as a fixed-dimensional vector through a Transformer encoder trained to predict both intra-tree relations and language classes.
The Euclidean distance between embeddings defines a metric over trees, and the Wasserstein distance is used to compare sets of trees.
Compared to classical tree distance methods such as Tree Edit Distance (TED) and PQ-grams, the proposed model achieves orders-of-magnitude faster computation while preserving TED-like properties.
The authors demonstrate applications to large-scale linguistic analysis across various languages, parser evaluation, and assessing the structural similarity of LLM-generated trees.
They also reveal empirical evidence that natural language trees are not context-free and visualize cross-linguistic clusters in the learned metric space.

**Strengths:**

-  The idea of constructing a learned metric space for sentence trees is conceptually new and elegant. It bridges structural linguistics and modern representation learning.

-  The proposed approach reduces the cubic cost of TED to linear time in tree size, enabling analysis of up to one million trees, a strong scalability result with practical implications.

- The experiments span 21 natural languages, random and context-free trees, and multiple LLM generations. This breadth convincingly demonstrates generality and interpretability.

- The paper covers broad related work, and overall writing is clear.

**Weaknesses:**

- My primary concern lies in the impact and benefit of the proposed approach to the broader community. Although the paper presents a technically sound method and Section 6 provides interesting analyses, such as parser evaluation, LLM syntax assessment, and linguistic typology visualization, these results mainly serve as illustrative examples rather than evidence of broader methodological or representational impact. The strong correlation with TED is interesting, but the paper does not clearly articulate why reproducing this metric is valuable for representation learning or how the resulting tree metric space could drive new progress in downstream tasks. A more elaborate discussion or empirical demonstration of potential applications where the learned metric brings further benefits would strengthen the paper.

- The authors compare with classical tree metrics (TED, tree kernels, PQ-grams) and tree-structured neural models (Tree-LSTM, Tree-Transformer), but omit modern graph embedding or graph neural network (GNN) baselines that could, in principle, also encode tree structures. While the authors justify this choice (line 98) by citing Zhang et al. (2024) that standard GNNs without explicit structural encodings fail to capture tree geometry,  this claim remains unverified in their own setting.
As ICLR emphasizes representation learning, it would be valuable to empirically demonstrate how standard GNNs fail to represent tree topology here, showing that the proposed model’s success arises from its representational inductive bias rather than from task-specific tuning.

**Questions:**

- How does the proposed tree metric yield broader representational or practical benefits beyond reproducing TED and illustrative analyses?

- Could the authors provide a more elaborated discussion or empirical analysis showing how standard GNNs fail to capture tree structure, clarifying whether the proposed model’s advantage stems from its inductive bias or specific design choices?

---

### Official Review · Reviewer_P7K3 · 2025-10-31

**Soundness:** 2
**Presentation:** 2
**Contribution:** 3
**Rating:** 2
**Confidence:** 3

**Summary:**

This paper focuses on the problem of tree metric space and proposes a novel method that builds a sentence tree metric space through representation learning of embedding trees in a vector space. It represents every sentence tree structure as a vector, with the Euclidean distance applied to construct the sentence tree metric. The experimental results show that the proposed method achieves lower time complexity than previous approaches and exhibits strong capability in predicting tree structures and classes.

**Strengths:**

1. The paper proposes a novel sentence trees representation learning approach by using deep learning to construct a metric space.

2. The proposed method demonstrates good efficiency and scalability.

**Weaknesses:**

1. The motivation of the proposed method should be strengthened.

2. The authors should enhance the expressiveness of the paper and avoid unclear or ambiguous descriptions.

3. The number of comparison methods is too limited.

**Questions:**

1. How does the time cost of the proposed method compare with that of TrLSTM and TrTransformer?

2. Considering the resources and time required to train the proposed method, is it computationally economical?

3. I would like to see further experimental analyses, such as the ablation study of distance function.

4. The authors should improve the quality of the figures. For example, Figure 1 contains red wavy lines, Figure 2 shows inconsistent font sizes and an unclear workflow, and in Figure 5, "Norwegian-Bokmaal" extends beyond the box.

5. The authors should cite more recent papers, as only 3/48 of the current references are from the past three years (2022).

---

### Official Review · Reviewer_2fux · 2025-11-02

**Soundness:** 2
**Presentation:** 1
**Contribution:** 2
**Rating:** 2
**Confidence:** 4

**Summary:**

This is a paper on embedding a tree structure of a sentence. While not using terminals, tree shapes are widely different from sentence to sentence and language and language, its accurate embedding is (if properly conducted) an important problem.
Although the method itself is quite vaguely represented and no guarantees for being a "metric" is presented, empirically the proposed embedding works better than various baselines and languages; however, how different these shapes actually are and how much these differences are reflected as a "metric" is not presented in this paper.

**Strengths:**

Tree shapes are of course important theme for natural language processing and linguistics, and their study through appropriate embeddings is also beneficial.
Tree shapes will be statistically different from language to language, thus experimental evaluations on many different languages and random trees (and LLM trees) are very interesting and important.

**Weaknesses:**

There are several drawbacks in the current form of this paper.

- First of all, the proposed embedding method is presented quite ambigously. There are many pages of results in this paper, but the central part of Section 3.2 is less than a page and without almost any mathematical explanation. Could you explain Figure 2 in words and equations? What is "cross-entropies of the positions of every node"? (i.e. are you using nonterminals associated with each node or completely concentrate on vanilla tree shapes?) There are no argument to regard this method of embedding better than other possible ways.
- Second, is the proposed method actually "metric"? There are no arguments that the resulting embeddings become a metric: just using Euclidean distance as a "metrtic" seems to be an overstatement. "Metric" also appears in the title, and you can be more careful for using that kind of mathematical statements.
- Finally, depth-first traversal also seems to be a heuristic. Since a tree structure can be represented as a binary matrix of connections (imagine CYK parsing), directly embedding that matrix seems to be more mathematically grounded way for embedding tree structures.

At least, this paper seems to need two pages for presenting the basic theory of embedding and its theoretical conclusions.

**Questions:**

See "Weaknesses".

---

### Official Review · Reviewer_hwzd · 2025-11-02

**Soundness:** 3
**Presentation:** 2
**Contribution:** 2
**Rating:** 4
**Confidence:** 4

**Summary:**

This paper introduces a novel approach to constructing a sentence tree metric space through representation learning of sentence structures. The proposed method embeds each sentence’s syntactic tree into a continuous vector space, where Euclidean distance is employed to define a metric over the sentence trees. The analysis further investigates the structural characteristics of natural language trees, comparing them not only across different languages but also against randomly generated trees.

**Strengths:**

The paper is supported by experimental evaluations that demonstrate the applicability of the proposed approach. The authors provide analyses that help clarify the relationship between linguistic structures and their learned representations.

**Weaknesses:**

Despite its merits, several issues should be addressed to improve the paper’s clarity and impact:

a. Limited applications of the proposed framework. The paper primarily focuses on tree-based representations of sentences. While a few application examples are provided, the practical utility of the sentence tree metric space remains somewhat restricted. Moreover, in natural language, many sentences exhibit incomplete or ambiguous syntactic structures, which may limit the proposed approach.

b. Insufficient comparison with classical NLP methods. Although various approaches exist for sentence representation in classical NLP, the experimental section includes only a small subset. Incorporating additional well-established methods would provide a more comprehensive evaluation and strengthen the empirical findings.

c. Lack of comparison with LLM-based sentence representations. Given the strong generalization capabilities of large language models (LLMs), they could serve as natural baselines for sentence representation—either through fine-tuning or prompt engineering. Including such comparisons would help contextualize the proposed method within the current NLP landscape.

d. Unclear definition of the tree metric space. In Section 3.2, the formal definition of the sentence tree metric space is vague. The paper lacks a rigorous mathematical specification of the metric and does not clearly demonstrate how the metric space properties are utilized in practical applications. Clarifying this section and providing theoretical justification would enhance the paper’s rigor.

**Questions:**

To strengthen the contribution, the authors are encouraged to (1) include additional baselines encompassing both classical NLP and LLM-based approaches, (2) explore broader or more realistic applications of the proposed tree metric space, and (3) refine the formal definition and theoretical grounding of the metric space.

---

### Note · Authors · 2025-11-13

I have read and agree with the venue's withdrawal policy on behalf of myself and my co-authors.